# Skin irritation testing using human iPSCs derived 3D skin equivalent model

Hyewon Shin[1☉], Se-Eun Kim[1], C-Yoon Kim[1], Suemin Lee[1], Ji-Heon Lee[2], Jieun Baek[2], Gujin Chung[2], Min Woo Kim[1], Jeong-Seop Oh[3], Shinhye Park[1], Yun Hyeong Lee[1], Youngin Jeong[1], Jeong Hwan Park[1], Yoonseo Kim[1], Myeonghee Lee[1], Seul-Gi Lee[1‡]*, Hyung Min Chung[4,5‡]*

1 College of Veterinary Medicine, Konkuk University, Gwangjin, Seoul, Republic of Korea, 2 R&D Center, CLECELL Inc., Beobwon-ro, Songpa-gu, Seoul, Korea, 3 Department of Veterinary Pathology, College of Veterinary Medicine, Seoul National University, Gwanak-Gu, Seoul, Republic of Korea, 4 Department of Stem Cell Biology, School of Medicine, Konkuk University, Gwangjin-Gu, Seoul, Republic of Korea, 5 Miracell Bio CO. Ltd., Gangnam-gu, Seoul, Korea

☉ This author contributed to this work as first author.
‡ These authors contributed equally to this work as corresponding authors, respectively.
* maxwisdom@konkuk.ac.kr (S-GL); hmchung@kku.ac.kr (HMC)

## Abstract

Artificial skin models have emerged as valuable tools for evaluating cosmetic ingredients and developing treatments for skin regeneration. Among them, 3D skin equivalent models (SKEs) using human primary skin cells are widely utilized and supported by standardized testing guidelines. However, primary cells face limitations such as restricted donor availability and challenges in conducting genotype-specific studies. To overcome these issues, recent approaches have focused on differentiating skin cells from human-induced pluripotent stem cells (hiPSCs). In this study, we developed a protocol to differentiate high-purity skin cells, such as fibroblasts (hFIBROs) and keratinocytes (hKERAs), from hiPSCs. To construct the hiPSC-derived SKE (hiPSC-SKE), a dermis was first formed by culturing a collagen and hFIBROs mixture within an insert. Subsequently, hKERAs were seeded onto the dermis, and keratinization was induced under air-liquid culture conditions to establish an epidermis. Histological analysis with hematoxylin and eosin staining confirmed that the hiPSC-SKE recapitulated the layered architecture of native human skin and expressed appropriate epidermal and dermal markers. Moreover, exposure to Triton X-100, a known skin irritant, led to marked epidermal damage and significantly reduced cell viability, validating the model's functional responsiveness. These findings indicate that the hiPSC-SKE model represents a promising alternative for various skin-related applications, including the replacement of animal testing.

**Data availability statement:** All relevant data are within the paper and its Supporting Information files.

**Funding:** This work was supported by the Industrial Strategic Technology Development Program (RS-2024-00448561) funded by the Ministry of Trade, Industry & Energy (MOTIE, Korea) in 2025. This paper was supported by the KU Research Professor Program of Konkuk University.

**Competing interests:** The authors have declared that no competing interests exist

## Introduction

The skin, as the body's outermost tissue, serves as a barrier against environmental stimuli and performs essential physiological functions [1,2]. It is primarily composed of the epidermis and dermis. The epidermis, the outermost layer of the skin, is mainly made up of keratinocytes and serves as a physical barrier protecting the body from external contaminants [1]. Beneath the epidermis lies the dermis, which is composed primarily of fibroblasts and collagen. This connective tissue provides elasticity and strength to the skin [2]. To replicate skin *in vitro*, 3D skin equivalent models (SKEs) have been developed using primary skin cells [3–5]. SKEs can be fabricated relatively easily using insert-based methods [6]. When keratinocytes are cultured on a surface exposed to air, they form the epidermis-like layer, creating an environment and structural complexity similar to that of human skin. The organization of fibroblasts within collagen helps maintain the elasticity and structure of SKEs, supported by the extracellular matrix (ECM) secreted by fibroblasts. Due to these features, SKEs are widely used in studies related to drug toxicity, disease modeling, and wound healing applications [5].

SKEs are widely used for skin irritation and corrosion testing, guided by international standards like the Organization for Economic Cooperation and Development test guideline 439 (OECD TG 439). This testing is crucial for evaluating the safety of cosmetics, pharmaceuticals, and chemicals, and OECD TG 439 promotes artificial skin models as a replacement for animal testing [7–9]. Recently, the U.S. Food and Drug Administration (FDA) introduced the FDA Modernization Act 2.0 in 2023, officially recognizing alternatives to animal testing in drug development and safety evaluation [10,11]. Similarly, the European Medicines Agency (EMA) announced that animal testing will be phased out in all product development processes starting in 2024. In response to these global trends, the use of SKEs has increased, with many studies addressing the limitations of existing models.

Artificial skin models used according to established guidelines typically include reconstructed human epidermis (RhE) products such as Episkin™, EpiDerm™, and SkinEthic™ RHE, as well as dermo-epidermal SKEs constructed from commercially available primary skin cells [7–9]. In addition to safety evaluations, skin cells are commonly used in research on skin diseases and therapeutic agents [6,12–15]. While primary cells reflect *in vivo* maturity, their use is limited by supply shortages, high costs, and senescence during passaging. Additionally, modeling genetic skin diseases is challenging due to difficulties in producing patient-specific products, and immune rejection complicates therapeutic transplantation. To address these issues, human induced pluripotent stem cells (hiPSCs) have emerged as a promising source [16,17]. Advantages of hiPSC-derived cells include: (1) unlimited supply through self-renewal, (2) disease modeling via patient-specific genetics or gene editing, and (3) improved therapeutic potential with better engraftment and low immunogenicity [16,17]. Several studies have successfully generated 3D SKEs from hiPSC-derived skin cells [18–21]. However, reports on hiPSC-derived SKEs (hiPSC-SKEs) remain limited, and a comprehensive study database is needed to optimize these models.

In this study, we established a protocol to differentiate skin cells from hiPSCs and generated hiPSC-SKEs. Morphological analyses confirmed that hiPSC-SKEs replicate human skin features, and skin irritation tests using Triton X-100 were successfully performed. These datasets are expected to contribute significantly to cosmetics, chemicals, and drug development as hiPSC-SKEs advance as skin models.

## Materials & methods

### Generation of human induced pluripotent stem cells (hiPSCs)

$1.5 \times 10^6$ human dermal fibroblasts (BJ cells; ATCC, VA, USA) were transfected with episomal vectors (pCLXE-hOCT4/p53, pCLXE-hSOX2/KLF4, and pCLXE-hL-MYC/LIN28A; Addgene, MA, USA) using the P2 Primary Cell 4D-Nucleofector Kit (Lonza, Basel, Switzerland) according to the manufacturer's protocol [22,23]. The mixture of cells and vectors was transferred into a Nucleocuvette Vessel and subjected to electroporation. Transfected cells were plated onto 1:100 Matrigel (Corning, NY, USA)-coated 6-well plates and cultured for 2 days in Dulbecco's Modified Eagle's Medium (DMEM; Gibco, NY, USA) supplemented with 10% fetal bovine serum (FBS; Sigma-Aldrich, St. Louis, MO, USA), 0.1 mM non-essential amino acids (NEAA; Gibco), and 1% penicillin/streptomycin (P/S; Gibco). After 2 days, the medium was replaced with TeSR-E7 medium (STEMCELL Technologies, Vancouver, Canada), which was changed daily. Cells were cultured for an additional 10–14 days until PSC-like colonies appeared. After picking the PSC-like colonies, they were seeded onto Matrigel-coated 4-well plates in mTeSR1 (STEMCELL Technologies Inc; Vancouver, BC, Canada) medium supplemented with 10 µM Y-27632 (Tocris Bioscience, Bristol, UK). The medium was changed daily with fresh mTeSR1 during subsequent culture.

### Generation of hiPSCs derived fibroblasts (hFIBROs)

hiPSCs were seeded onto 1:100 Matrigel-coated 60 mm cell culture dishes and maintained in mTeSR1 medium with daily medium changes. Upon reaching 90–100% confluency after 3 days, the cells were gently dissociated using DPBS (Welgene, Gyeongsan, South Korea) containing 0.5 mM EDTA (Gibco). The detached hiPSCs were resuspended in 6 ml mTeSR1 supplemented with 10 µM Y-27632 and transferred to F127-coated (Sigma-Aldrich) 60 mm Petri dishes to initiate embryoid body (EB) formation (D0). Cells were cultured on a shaker at 65–70 rpm in a 37 °C incubator with 5% $CO_2$ for 24 hr. On the following day (D1), EB formation was confirmed, and mTeSR1 was replaced daily for two additional days (until D3). Approximately 150–200 EBs were then transferred onto Matrigel-coated 60 mm culture dishes and cultured in fibroblast differentiation medium 1 (FDM1) for 3 days (D3-D6). FDM1 consisted of a 3:1 mixture of DMEM (Gibco) and F12 (Gibco), supplemented with 5% FBS, 5 µg/ml insulin (Sigma-Aldrich), and 10 ng/ml epidermal growth factor (EGF; PeproTech, NJ, USA). After 3 days, the medium was changed to fibroblast differentiation medium 2 (FDM2), which is FDM1 supplemented with 25 ng/ml bone morphogenetic protein 4 (BMP4; PeproTech), and cultured for another 3 days (D6-D9). Subsequently, the cells were cultured in fibroblast differentiation medium 3 (FDM3) for 7 days (D9-D16). FDM3 consisted of a 1:1 mixture of DMEM and F12 supplemented with 5% FBS and 1% NEAA. Upon reaching 70–80% confluency (Day 16), cells were dissociated using a 5:1 mixture of TrypLE (Gibco) and 0.25% Trypsin-EDTA (Gibco). The detached cells were replated on Matrigel-coated dishes in FDM1 and maintained until Day 30 (D30) (Fig 1A).

### Generation of hiPSCs derived keratinocytes (hKERAs)

70−80 EBs were seeded onto Matrigel-coated 60 mm cell culture dishes and cultured in keratinocyte differentiation medium 1 (KDM1) for 7 days, with daily medium changes (D3-D10). KDM1 consisted of a 3:1 mixture of DMEM and F12, supplemented with 2% FBS, 5 µg/mL insulin, 25 ng/ml EGF, 25 ng/ml BMP4, and 1 µg/ml retinoic acid (RA; Sigma-Aldrich). After 7 days, the medium was replaced with keratinocyte differentiation medium 2 (KDM2) and cells were cultured for an additional 7 days (D10-D17). KDM2 consisted of Defined Keratinocyte Serum-Free Medium (DKSFM;

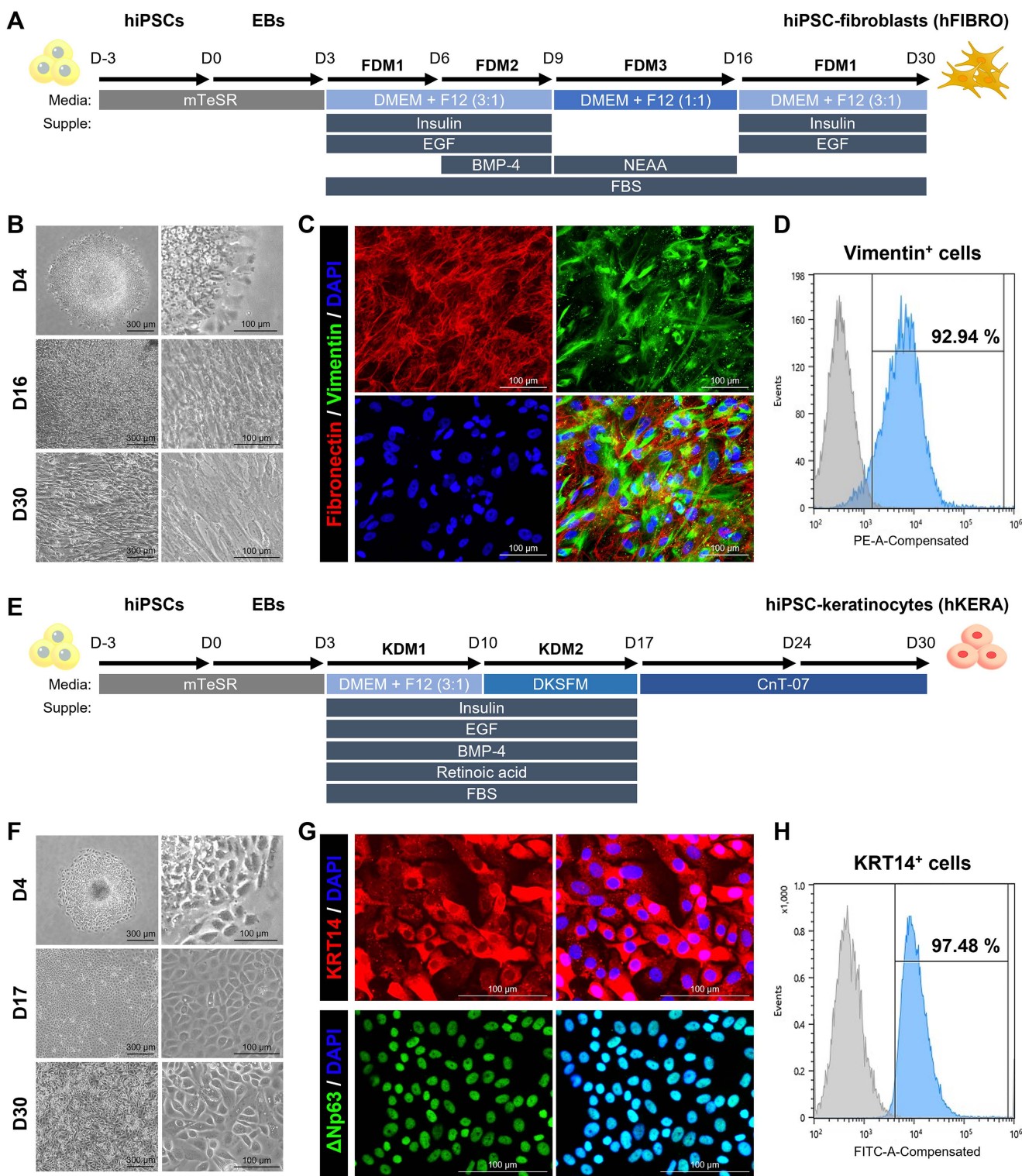

**Fig 1. Generation of hFIBROs and hKERAs.** (A) Differentiation protocol of hFIBROs. (B) The morphology of hFIBROs at each day (Day 4, 16 and 30). Left scale bar: 300 μm and right scale bar: 100 μm. (C) IF staining for fibronectin and vimentin. Scale bar: 100 μm. (D) FACS for Vimentin⁺ cells in hFIBROs. (E) Differentiation protocol of hKERAs. (F) The morphology of hKERAs at each day (Day 4, 17 and 30). Left scale bar: 300 μm and right scale bar: 100 μm. (G) IF staining for KRT14 and ΔNp63. Scale bar: 100 μm. (H) FACS for KRT14⁺ cells in hKERAs.

Gibco) supplemented with 0.5% FBS, 5 µg/ml insulin, 20 ng/ml EGF, 20 ng/ml BMP4, and 1 µg/ml RA. Subsequently, the cells were cultured for 7 days in CnT-07 medium (CELLnTEC, Bern, Switzerland) (D17-D24). When cells reached 70–80% confluency (D24), they were dissociated using a 5:1 mixture of TrypLE (Gibco) and 0.25% Trypsin-EDTA (Gibco). The detached cells were replated on Matrigel-coated culture dishes and maintained in CnT-07 medium until Day 30 (D30) (Fig 1E).

### Generation of human iPSCs derived 3D Skin Equivalent Model

To construct the dermal layer, a mixture of 4.5 mg/ml collagen type I (Corning, NY, USA, Cat# 354249) and $6 \times 10^4$ hFI-BROs was poured into a porous polyester membrane insert (Corning) and allowed to solidify in a 37 °C incubator with 5% $CO_2$ for 30 min. Then, the construct was cultured in FDM1 medium with daily medium changes for 3 days (D0-D3). Subsequently, $1.5 \times 10^6$ hKERAs were seeded onto the surface of the dermal layer in CnT-07 medium, which was replaced daily for 3 days (D3-D6). The culture was then switched to CnT-airlift medium (CELLnTEC) supplemented with 1 mM calcium chloride ($CaCl_2$; PromoCell, Heidelberg, Germany) and maintained for an additional 4 days (D6-D10). To induce epidermal keratinization, the medium was completely removed from the insert, and an air–liquid culture was maintained for 20 days (D10-D30). CnT-airlift medium was added beneath the insert and replaced every other day [24].

### Flow cytometry analysis (FACS)

hFIBROs and hKERAs were fixed in 4% Paraformaldehyde solution (PFA; Biosesang, Yongin, Korea) for 20 min and resuspended in FACS solution after washing. Then, fixed hFIBROs and hKERAs were stained each using PE-conjugated Mouse Anti-Human Vimentin (BD Biosciences; CA, USA, Cat# 562337; 1:25) and FITC-conjugated Anti-Cytokeratin 14 antibody (KRT14; abcam, Cambridge, England, Cat# ab77684; 1:25) for 30 min at 4 °C. To confirm Vimentin and KRT14 expression rate, analysis was performed using FACS Calibur and Cell Quest software (BD Biosciences).

### Immunofluorescence analysis

hFIBROs and hKERAs were fixed with 4% PFA for 25 min at 4 °C and replaced with PBS. The preparation was blocked and permeabilized with blocking solution (PBS containing 0.1% Triton X100 and 3% normal goat serum) on shaker for 30 min at room temperature (RT). Next, it was treated to the primary antibodies diluted in blocking solution overnight (O/N) at 4 °C. As primary antibodies, anti-Vimentin (abcam, Cat# ab8978; 1:200), anti-Fibronectin (abcam, Cat# ab2413; 1:200), anti-ΔNp63 (abcam, Cat#203826; 1:100), and anti-keratin 14 (KRT14; abcam, Cat# ab181595; 1:200) were used. After O/N, it was washed twice with PBS for 15 min each and treated with secondary antibodies diluted in blocking solution for 2 hr under light protection. As secondary antibodies, Alexa 488 goat anti-mouse IgG (Invitrogen, Grand Island, NY), Alexa 488 goat anti-rabbit IgG (Invitrogen) and Alexa 594 goat anti-rabbit IgG (Invitrogen) were used for 1:700. Next, it was washed twice with PBS for 15 min each and treated with DAPI (4′,6-diamidino-2-phenylindole; Sigma-Aldrich, Cat# D9542) for 1 min. DAPI was used as the nuclear counterstain.

hiPSC-SKEs were fixed with 4% PFA on D30 at 4 °C O/N. The fixed samples were dehydrated in graded sucrose (15% to 30%) for 2 days and planted in OCT compound (Sakura Finetek, Tokyo, Japan) for cryosection. Cryosections were sliced by sectioning them into about 8 µm size on a cryotome. The sections were blocked and permeabilized with blocking solution for 30 min at RT. Subsequently, they were treated to the primary antibodies diluted in blocking solution O/N at 4 °C in wet chamber. As primary antibodies, anti-Vimentin (abcam, Cat# ab8978; 1:200), anti-Fibronectin (abcam, Cat# ab2413; 1:200), anti-ΔNp63 (abcam, Cat#203826; 1:100), anti-KRT14 (abcam, Cat# ab181595; 1:200), anti-platelet-derived growth factor receptor α (PDGFRα; abcam, ab32570), anti-KRT10 (abcam, Cat# ab76318; 1:200), and anti-Loricirin (abcam, Cat# ab198994; 1:100) were used. Next day, they were washed 3–5 times with PBS and treated with secondary antibodies diluted in blocking solution for 2 hr at RT under light protection. As secondary antibodies, Alexa 488 goat anti-mouse IgG, Alexa 488 goat anti-rabbit IgG, and Alexa 594 goat anti-rabbit IgG were used for 1:700. Then,

they were washed 3 times with PBS and treated with DAPI. After that, they were washed 3 times with PBS and mounted through mounting solution (Vector labs, CA, USA). The fluorescent dye color of the images was changed for visual effect.

All images were analyzed by using a fluorescence microscope, Nikon TE2000-U (Nikon, Tokyo, Japan).

### Gel electrophoresis

To analyze the gene expression in hFIBROs and hKERAs, cells were harvested on day 30 of differentiation. Total RNA was isolated using TRIzol reagent (Invitrogen, Grand Island, NY, USA) following the manufacturer's instructions. 1 µg RNA was used for cDNA synthesis with AccuPower RT PreMix and Oligo(dT) primers (Bioneer, Daejeon, Korea). The synthesized cDNA was then mixed with gene-specific primers and AccuPower® PCR PreMix (S1 Table in S1 File). PCR was performed and the presence of target gene bands was verified for each sample by electrophoresis. The information of primers was described in S1 Table in S1 File. Also, we used GAPDH as PCR control.

### Histological analysis

The fixed samples for paraffin block were subjected to tissue processing and paraffin embedding. Paraffin sections were prepared by sectioning at 5 µm through a microtome. Hematoxylin and Eosin (H&E) staining was performed in Hematoxylin for 6 min and washed 3 times in 0.5% HCl-ethanol. Subsequently, they were neutralized with 0.5% Ammonia solution, and stained in Eosin Y for 7 min. After that, they were dehydrated and mounted. Human circum skin tissue and mouse dorsal skin tissue were sampled as native skin.

### Skin irritation testing

Cell Counting Kit-8 (CCK-8; Dojindo, Kumamoto, Japan) was used to measure cell viability. 300 µl CCK-8 working solution (a 1:9 mixture of CCK-8 reagent and CnT-airlift medium) was added to the insert containing D30 hiPSC-SKE and the baseline absorbance was measured. After incubation in 37 °C, 5% $CO_2$ incubator for 2 hr, 100 µl of the solution in the insert was transferred to flat bottom 96 well plate and measure the baseline absorbance. Then, the solution in the insert was replaced with fresh media and incubated for 30 min. After washing, 3% Triton X-100 diluted in medium was treated in the drug treatment group and incubated for 1.5 hr. Next, it was replaced with 300 µl CCK-8 working solution and incubated for 2 hr. After incubation, 100 µl of the medium in the insert was transferred to a 96 well plate and measure the absorbance. The absorbance was measured at a wavelength of 450 nm using an Epoch microplate reader (Agilent, CA, USA).

### Statistical analysis

All experiments were performed at least three times. Statistical analyses were performed using the GraphPad Prism software (La Jolla, CA, USA; Version 10). Data were presented as mean ± SEM, and the statistical significance of the experimental results was calculated using either a T-test or one-way ANOVA with Tukey's method for comparison between groups. A p value (*$p < 0.05$, **$p < 0.01$ and ***$p < 0.001$) was considered statistically significant.

## Results

### Differentiation and characterization of hFIBROs and hKERAs

To generate human induced pluripotent stem cells (hiPSCs) derived fibroblasts (hFIBROs), we established a protocol based on cytokines and small molecules by referring to previous reports (Fig 1A) [21,25]. During differentiation, morphological changes consistent with mesenchymal transition were observed. Embryoid bodies (EBs) plated onto Matrigel-coated dishes proliferated and began adopting a spindle-shaped, fibroblast-like morphology by day 16 (D16), which was maintained after passaging up to day 30 (D30) (Fig 1B). IF analysis was performed for vimentin, a fibroblast marker [26], and fibronectin, an extracellular matrix (ECM) protein secreted by fibroblasts [27]. Most of the differentiated

cells (DAPI) expressed vimentin, and fibronectin was secreted beneath the vimentin⁺ cells (Fig 1C). The purity of hFIBROs was assessed by analyzing the density of vimentin⁺ cells per field, which was 97.01±0.73% (S1A Fig in S1 File). Also, FACS analysis confirmed that 92.94% of the cells were vimentin⁺, consistent with IF staining. (Fig 1D). Next, we established a protocol for generating hiPSC derived keratinocytes (hKERAs) (Fig 1E) [21,24,25]. Following EB attachment, cells began to exhibit a cobblestone morphology characteristic of keratinocytes by day 17 (D17), which persisted through D30 (Fig 1F). To confirm the expression of keratinocyte-specific markers in hKERAs, we performed staining for KRT14 and ΔNp63. KRT14 is a type I intermediate filament protein predominantly expressed in the basal keratinocytes of the epidermis, while ΔNp63 is a transcription factor involved in the regulation of early keratinocyte differentiation and proliferation [28–30]. Most of the differentiated cells (DAPIˉ) expressed ΔNp63 and KRT14 (Fig 1G). The purity of hKERAs was assessed by analyzing the density of ΔNp63⁺ cells per field, which was 98.09±0.24% (S1B Fig in S1 File). FACS analysis further confirmed the efficiency of differentiation, with 97.48% of cells expressing KRT14 (Fig 1H). To further confirm the absence of residual undifferentiated cells in the differentiated hFIBROs and hKERAs, gel electrophoresis analysis for pluripotency markers was performed (S2 Fig in S1 File). Pluripotent markers such as SOX2 and OCT4 were strongly expressed in undifferentiated hiPSCs but were undetectable in both hFIBROs and hKERAs. In contrast, fibroblast-specific markers (PDGFRα, Col3A1, and fibronectin) were exclusively expressed in hFIBROs (S2A Fig in S1 File), while keratinocyte-specific markers (KRT14 and ΔNp63) were detected only in hKERAs (S2B Fig in S1 File). These results confirm successful lineage-specific differentiation and effective elimination of PSCs, demonstrating that the established protocols reliably generate hFIBROs and hKERAs.

## Generation of hiPSC-SKE

To fabricate skin equivalents using hiPSC derived skin cells (hiPSC-SKE), a previously reported method based on insert and collagen was employed (Fig 2A) [6,12,25]. After inserting a mixture of hFIBROs and collagen into the insert, the dermis was formed through gelation and subsequent culture (Fig 2A and 2B). Subsequently, hKERAs were seeded onto the constructed dermis to form an epidermal layer, and keratinization was induced via air-liquid (ALI) culture (Fig 2A and 2B). As differentiation progressed, the upper surface of hiPSC-SKE exhibited changes. After ALI culture, the epidermal layer became uneven, with some edges contracting. (Fig 2B). H&E staining was performed to evaluate epidermal morphology at early (D20; ALI culture for 10 days) and late (D30; ALI culture for 20 days) stages of ALI culture (Fig 2C). Keratinization of the epidermis was more prominent at ALI-20 days than at ALI-10 days of ALI culture. The epidermal thickness was significantly increased at ALI-20 days (ALI-10 days: 38.82±2.20 μm vs. ALI-20 days: 63.62±8.06 μm), accompanied by thickening of the stratum basale (SB) and stratum spinosum (SS)-like layers. Furthermore, a stratum corneum (SC)-like structure was observed at the air-exposed surface at ALI-20 days (Fig 2C and 2D) [13]. Comparative H&E analysis revealed that both the dermal and epidermal layers of hiPSC-SKE closely resemble those of native human and mouse skin in structure (Fig 2E).

## Characterization of Cellular and ECM Distribution in hiPSC-SKE

To investigate the progression of keratinization induced by ALI culture in the epidermis, representative markers of major epidermal layers were examined at 10 and 20 days of ALI culture. IF staining for KRT14 and ΔNp63, representative markers of keratinocytes predominantly expressed in the SB, showed strong expression in the lower part of the epidermis (Fig 3A and 3B). Notably, ΔNp63 expression gradually decreased toward the upper layers at ALI-20 days (D30), compared to ALI-10 days (D20) (Fig 3B). KRT10, a marker primarily expressed in the SS, was strongly detected above the SB, and its expression domain became thicker at ALI-20 days compared to ALI-10 days (Fig 3C). Furthermore, loricrin, a late differentiation marker expressed in the SC, was not detected at ALI-10 days but was clearly expressed in a band-like pattern at ALI-20 days (Fig 4D) [20,28,31]. Next, the distribution of hFIBROs within the dermis was assessed using the

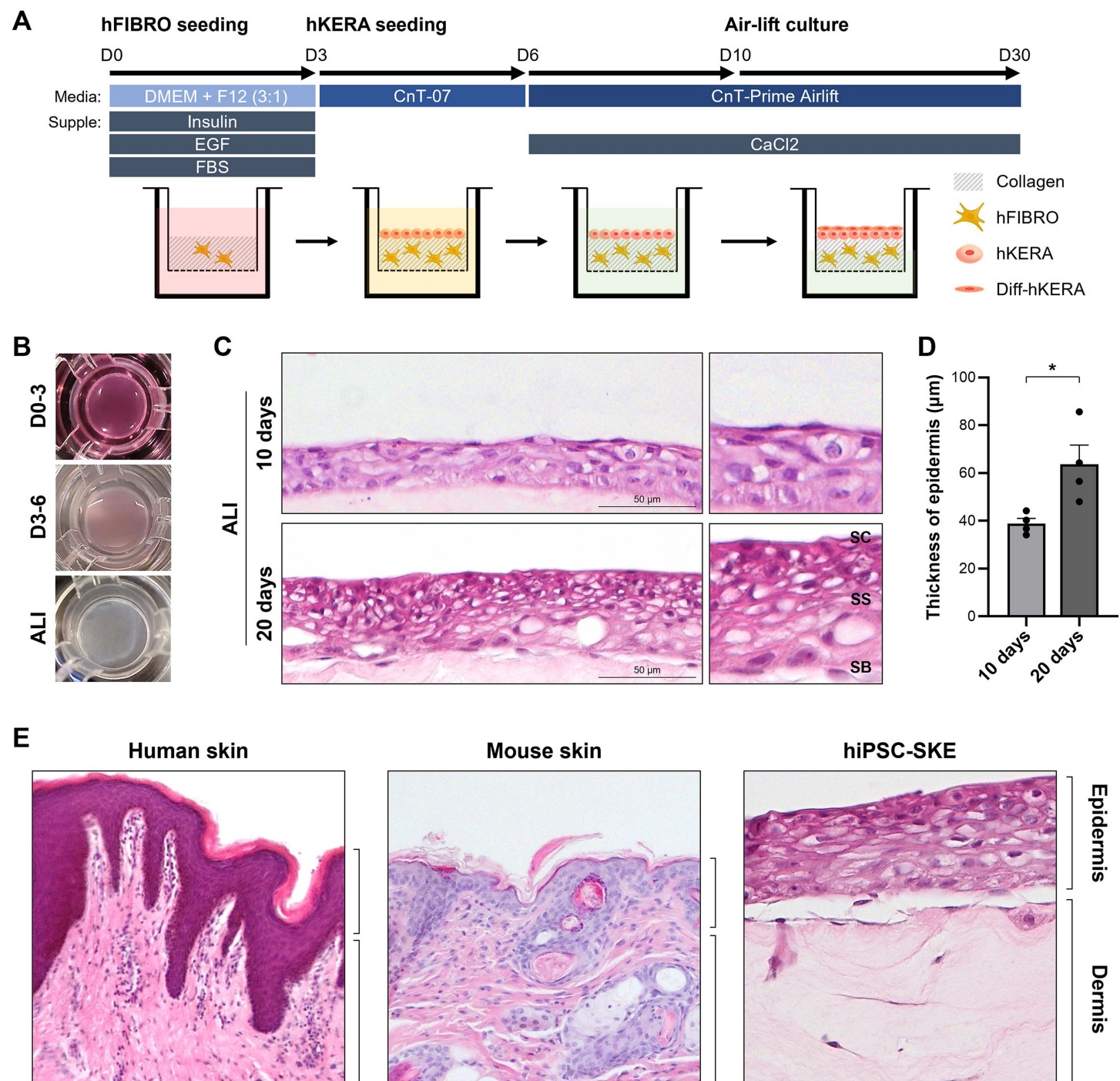

**Fig 2. Generation of hiPSC-SKE.** (A) Protocol for generation of hiPSC-SKE. (B) Images of hiPSC-SKE in the inserts at each differentiation stage. (C) Comparison of epidermis formation according to ALI culture duration (10 and 20 days) through H&E staining. SC: stratum corneum. SG: stratum granulosum. SB: stratum basale. Scale bar: 50 μm. (D) Comparison of epidermis thickness according to ALI culture duration (10 and 20 days) (n=4). (E) Morphology between native skin (human and mouse) and hiPSC-SKE through H&E staining. Scale bar: 25 and 50 μm. Yellow dotted boxes: enlarged images. Data are presented as mean values±SEM. $p$ value: *$p<0.05$.

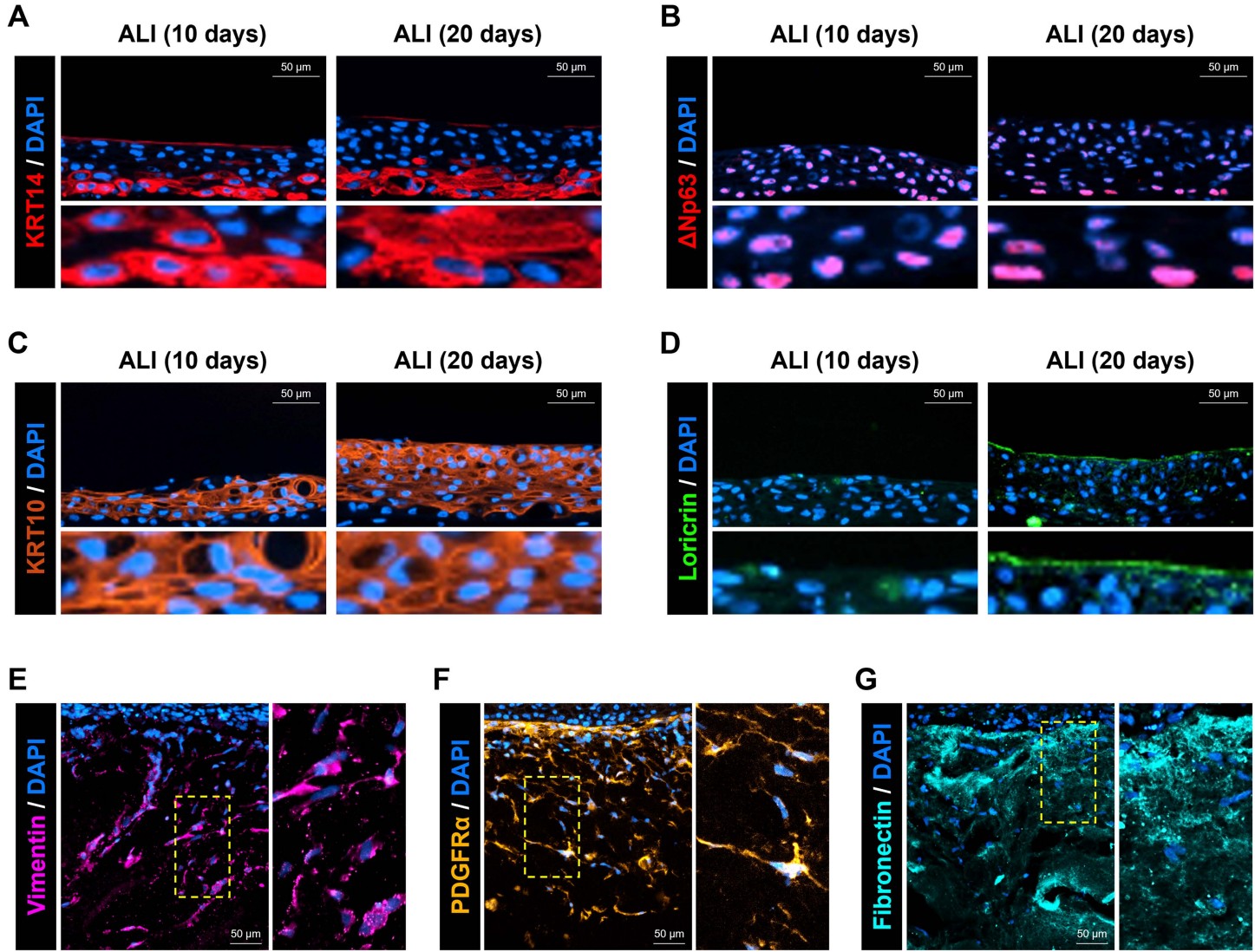

**Fig 3. Verification of skin related markers in hiPSC-SKE.** IF staining for (A) KRT14 and (B) ΔNp63 identifying SB distribution in hiPSC-SKE according to ALI culture duration (10 and 20 days). IF staining for (C) KRT10 to identify SS distribution and (D) Loricrin to identify SC distribution. IF staining for (E) vimentin, (F) PDGFRα, and (G) fibronectin to identify hFIBROs in hiPSC-SKE at D30 (ALI culture for 20 days). Scale bar: 50 µm. Yellow dotted boxes: enlarged images.

fibroblast-specific markers Vimentin and PDGFRα. Vimentin⁺ and PDGFRα⁺ cells exhibited mesenchymal morphology and were distributed within the collagen-containing dermis (Fig 3E and 3F). Fibronectin, an ECM component secreted by fibroblasts, was expressed in the dermis but not in the epidermis. Notably, fibronectin showed strong expression in the dermal region adjacent to the epidermis (Fig 3G). These results demonstrated that the application of hFIBROs and hKERAs can sufficiently implement artificial skin that can reflect the main skin constituent layers and the keratinization of epidermis.

## Skin irritation testing using hiPSC-SKE

Skin irritation testing is a type of alternative animal testing that evaluates cell damage in artificial skin models following exposure to chemical substances (Fig 4A). In this test, sodium dodecyl sulfate (SDS) or Triton X-100, which are known to

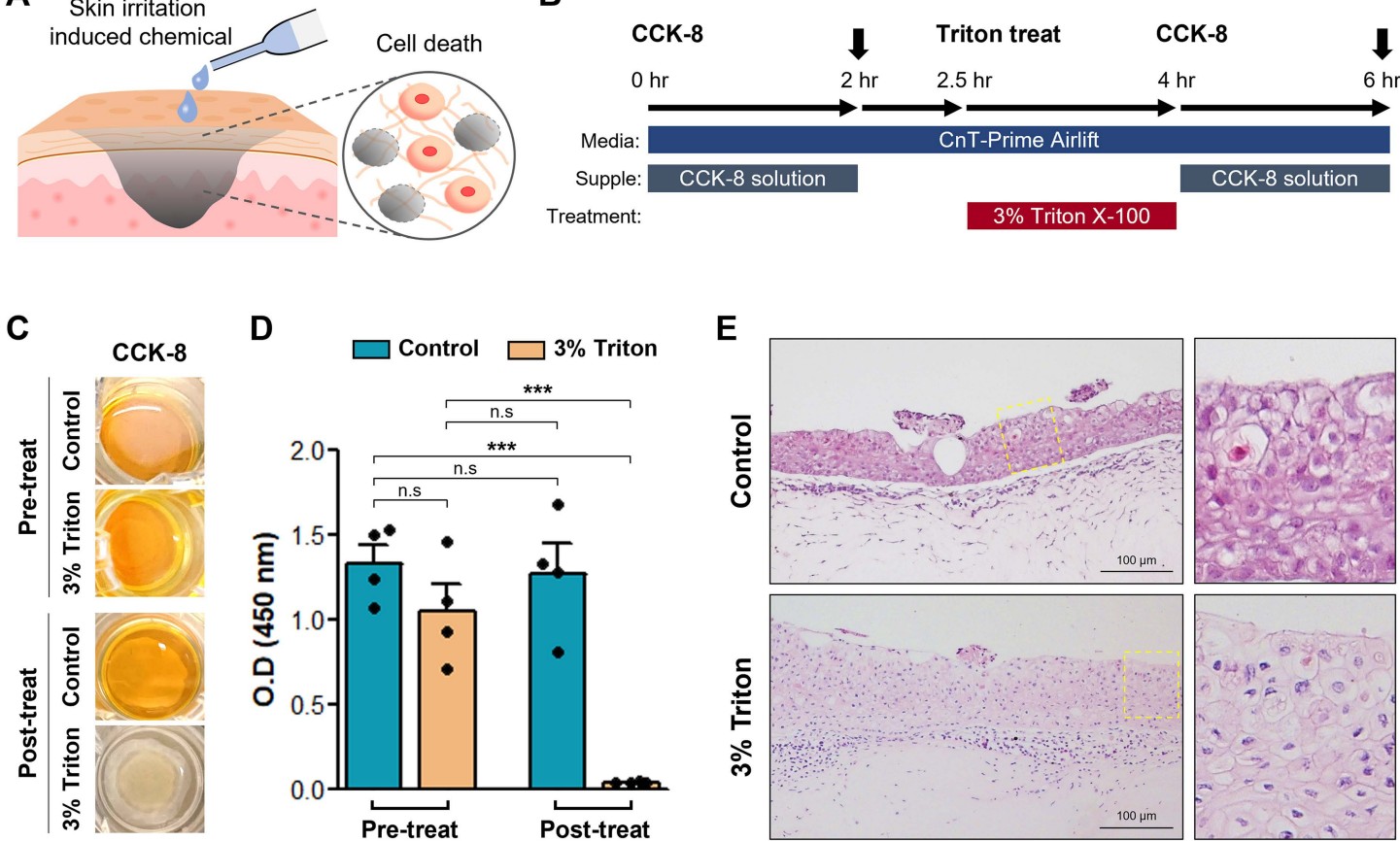

**Fig 4. Skin irritation testing using hiPSC-SKEs.** (A) Schematic diagram of skin irritation induced chemical stimulation. (B) Protocol of the skin irritation testing by CCK-8 assay. Arrow: O.D value measurement time. (C) Comparison of color changes before (Pre-treat) and after treatment (Post-treat) with the control group and 3% Triton using CCK-8 solution. (D) Changes in O.D value of CTL and 3% Triton-treated groups (n = 4). Data are presented as mean values ± SEM. n.s: not significant. *p* value: ***p < 0.0001. (E) Comparison of morphology between the control and 3% Triton using H&E staining. Scale bar: 100 μm. Yellow dotted boxes: enlarged images.

cause cytotoxicity as a side effect, are used as a positive control, and the degree of tissue damage is assessed through an absorbance-based cell viability test [32]. To evaluate the utility of hiPSC-SKEs in skin irritation testing, a CCK-8 based cell viability assay using 3% Triton X-100 was conducted. Triton X-100 is a nonionic detergent that disrupts cell membranes by specifically binding to membrane lipids, damaging the structural integrity of the membrane. This increases membrane permeability and causes ion imbalance, ultimately disrupting cellular homeostasis and inducing cell death. Additionally, increased cellular stress damages the mitochondrial membrane potential, further promoting cell death [33–36]. Triton X-100 induces reproducible skin irritation under standardized conditions with minimal nonspecific cytotoxicity, making it a widely used positive control in skin irritation testing. Prior to treatment with 3% Triton, the baseline (pre-treatment; pre-treat) optical density (O.D.) of each hiPSC-SKE was measured using the CCK-8 assay. To ensure proper penetration of the treatment agent, 3% Triton was applied inside the insert containing the hiPSC-SKEs, while only basic medium was added to the outer well. As a negative control, culture medium (CnT-Prime Airlift) was used. After 1.5 hr of treatment, the hiPSC-SKEs were washed, and the post-treatment (post-treat) O.D. was measured again using the CCK-8 assay (Fig 4B). In this assay, WST-8 is reduced by mitochondrial dehydrogenase to produce orange-colored

formazan [37]. As shown in Fig 4C, similar color changes were observed between the control and 3% Triton groups at the pre-treat stage, with no significant differences in O.D. (Fig 4D). However, after post-treat stage, the control group showed no change in color, whereas the 3% Triton group exhibited only a faint orange color (Fig 4C). Correspondingly, a significant decrease in O.D. was observed in the 3% Triton group (pre-treat: 1.053 ± 0.158; post-treat: 0.045 ± 0.002), while the control group maintained relatively stable O.D. (pre-treat: 1.336 ± 0.109; post-treat: 1.272 ± 0.179) (Fig 4D). These results suggest that 3% Triton induced cell death in hiPSC-SKEs, leading to reduced viability. Morphological analysis using H&E staining further confirmed severe damage to the epidermal layer near the air interface in the 3% Triton group, which was not observed in the control group (Fig 4E). In summary, hiPSC-SKE has been proven to be a valuable model for skin safety assessment of various skin irritation.

## Discussion

Our study demonstrated that hiPSC-derived skin equivalents (hiPSC-SKEs) can serve as a viable alternative to conventional primary skin cell-based models. hiPSC-SKE was generated by differentiating hiPSC-derived fibroblasts and keratinocytes. The formation of skin composition layers resembling native skin was confirmed, along with the cellular distribution within each layer. Furthermore, treatment of hiPSC-SKEs with 3% Triton X-100, followed by assessments of cell viability and morphological changes, supported their potential utility in skin irritation testing.

Previous studies have reported the development of SKEs using hiPSC-derived skin cells as substitutes for primary skin cells. Some of these models incorporated a mixture of hiPSC-derived and primary skin cells [18,20], while others were constructed entirely from hiPSC-derived fibroblasts and keratinocytes [18,19,21]. When compared to previous reports, our SKEs showed morphological differences—such as variations in epidermal and SC thickness and dermal cell distribution. These differences may result from factors such as cell density, culture period, and media composition. Despite these variations, most SKEs, including ours, exhibited structural patterns comparable to native human skin. Epidermal maturation in native skin proceeds through keratinization, starting from the SB and forming the SS, SG, and SC toward the air interface. In the SB, proliferating keratinocytes predominantly express KRT14 and ΔNp63, as reported in native skin [18–21]. Consistent with this, our results confirmed that KRT14 and ΔNp63 were strongly expressed in the basal region adjacent to the dermis in hiPSC-SKEs generated by ALI culture, with expression levels gradually diminishing toward the surface (Fig 3A and 3B) [38]. In native skin, KRT1 and KRT10 are primarily expressed in the SS, while loricrin and filaggrin are mainly expressed in the SC [39]. In our hiPSC-SKEs, we also demonstrated that the KRT10+ layer thickened as ALI culture progressed, and loricrin, which was not observed at ALI-10 days, became detectable by ALI-20 days (Fig 3C and 3D). In the dermis, fibroblasts exhibit a spindle-shaped morphology typical of mesenchymal cells [15]. Similarly, our IF analyses for vimentin and PDGFRα demonstrated that fibroblasts in hiPSC-SKE were distributed in a spindle shape (Fig 3C and 3D). The epidermis-dermis junction, also known as the BM, connects the dermis to the SB layer of the epidermis [40]. Type IV collagen is the major structural component of the BM, forming a meshwork that incorporates laminin and fibronectin [40]. Fibronectin contributes to basal lamina synthesis and promotes keratinocyte regeneration in the epidermis by binding to the collagen basal lamina [41]. IF analysis revealed that fibronectin was expressed in the dermal layer of hiPSC-SKE, with particularly high expression in the BM region (Fig 3E).

Skin irritants penetrate the SC and damage basal skin cells, triggering an inflammatory response in the dermis that leads to erythema and edema, which are key signs of skin irritation. To evaluate this *in vitro*, artificial skin models have been developed as alternatives to animal testing, supported by a robust database that now serves as a guideline. Skin irritation testing using SKEs typically involves cell viability assays, such as the MTT assay, after topical application of test substances [7–9]. According to the United Nations Globally Harmonized System of Classification and Labeling of Chemicals (UN GHS), a substance is classified as non-irritant if the average tissue viability after treatment exceeds 50%. In contrast, substances resulting in ≤50% viability are classified as irritants, and further assessment for skin corrosivity is required for detailed categorization. In this study, the % change in post-treat O.D. values relative to pre-treat values

was analyzed for each group. The viability rate of hiPSC-SKEs was 94.206 ± 8.371% in the control group, compared to 4.621 ± 0.738% in the group treated with 3% Triton (Fig 4D). These results confirm that Triton X-100, a well-established positive control for skin irritation in primary cell-based SKE models, also induces significant irritation in hiPSC-SKEs.

hiPSC-SKEs demonstrated marker expression and irritation responses comparable to primary skin cell-based SKEs, supporting their potential as reliable alternatives. hiPSCs offer advantages such as overcoming limitations in cell supply, cost, and donor variability. However, a key challenge remains as hiPSC-derived cells often exhibit an immature phenotype [42,43]. Similarly, hiPSC-derived skin organoids lack structural completeness, such as Collagen VII and immune or endothelial cells [44,45]. This immaturity may also affect hiPSC-derived skin cells, potentially explaining the need for ALI culture durations exceeding 20 days. Moreover, SKEs generated from primary skin cells often display a thicker SC compared to hiPSC-SKEs [20,24]. To enhance model maturity, future studies should incorporate additional skin constituent cells to better reflect *in vivo* conditions. Batch-to-batch variability during hiPSC differentiation also poses challenges for reproducibility. To address this, we are exploring previously reported droplet-based bioprinting and nebulization technologies to standardize hiPSC-SKE fabrication. In summary, the hiPSC-SKE model offers a scalable, physiologically relevant platform with promising applications in safety testing, disease modeling, and regenerative medicine.

## Supporting information

**S1 File. S1 Table. Infromation of primers.** Genes: GAPDH (housekeeping gene for control), puripotency marekrs (OCT4 and SOX2), fibroblast-specific markers (PDGFRα, Col3A1, and Fibronectin), keratinocyte-specific markers (KRT14 and ΔNp63). **S1 Fig. IF staining for purity check in hFIBRO and hKERA.** (A) Staining for vimentin, a fibroblast-specific marker, in hFIBRO and distribution of vimentin+ cells per fields (n = 3). (B) Staining for ΔNp63, a keratinocyte-

specific marker, in hKERA and distribution of ΔNp63 + cells per fields (n = 3). White arrows: DAPI where each marker is not expressed. Scale bar: 100 μm. **S2 Fig. Electrophoresis analysis for pluripotency and skin cell-specific markers by hiPSC, hFIBRO, and hKERA.** Genes: GAPDH (housekeeping gene for control), puripotency marekrs (OCT4 and SOX2), fibroblast-specific markers (PDGFRα, Col3A1, and Fibronectin), keratinocyte-specific markers (KRT14 and ΔNp63).
(DOCX)

## Acknowledgments

We thank all members of the team for their great support.

## Author contributions

**Conceptualization:** Hyewon Shin, Seul-Gi Lee, Hyung Min Chung.

**Data curation:** Hyewon Shin, Se-Eun Kim, C-Yoon Kim, Suemin Lee, Ji-Heon Lee, Jieun Baek, Min Woo Kim, Jeong-Seop Oh, Shinhye Park, Yun Hyeong Lee, Youngin Jeong, Jeong Hwan Park, Yoonseo Kim, Myeonghee Lee, Seul-Gi Lee.

**Formal analysis:** Hyewon Shin, Se-Eun Kim, C-Yoon Kim, Suemin Lee, Ji-Heon Lee, Jieun Baek, Gujin Chung, Min Woo Kim, Jeong-Seop Oh, Shinhye Park, Yun Hyeong Lee, Youngin Jeong, Jeong Hwan Park, Yoonseo Kim, Myeonghee Lee, Seul-Gi Lee, Hyung Min Chung.

**Project administration:** Hyewon Shin, Seul-Gi Lee, Hyung Min Chung.

**Supervision:** Seul-Gi Lee, Hyung Min Chung.

**Writing – original draft:** Hyewon Shin, Seul-Gi Lee, Hyung Min Chung.

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
