## [Decision Letter · Decision Letter 0]

25 Mar 2025

PONE-D-25-01505Skin Irritation Testing using Human iPSCs derived 3D Skin Equivalent ModelPLOS ONE

Dear Dr. Chung,

Thank you for submitting your manuscript to PLOS ONE. After careful consideration, we feel that it has merit but does not fully meet PLOS ONE’s publication criteria as it currently stands. Therefore, we invite you to submit a revised version of the manuscript that addresses the points raised during the review process.

1) I am not confident that we have a good quality of 3D skin. As pointed out by one of the reviewers, "There is no stratum granulosum and no stratum corneum formed. Differences in stratum basale and spinosum in H&E staining are only minor";2) Please, see the comments raised by both the reviewers.

We look forward to receiving your revised manuscript.

Kind regards,

Paulo Lee Ho, Ph.D.

Academic Editor

PLOS ONE

Journal Requirements:

Reviewers' comments:

Reviewer's Responses to Questions

**Comments to the Author**

1. Is the manuscript technically sound, and do the data support the conclusions?

Reviewer #1: Partly

Reviewer #2: Partly

2. Has the statistical analysis been performed appropriately and rigorously? 

Reviewer #1: Yes

Reviewer #2: N/A

3. Have the authors made all data underlying the findings in their manuscript fully available?

Reviewer #1: Yes

Reviewer #2: Yes

4. Is the manuscript presented in an intelligible fashion and written in standard English?

Reviewer #1: Yes

Reviewer #2: No

5. Review Comments to the Author

Reviewer #1: All comments are being submitted in a separate file in order to respect the limit of characters.

Please find the attached document for answer the required questions. I don't have any concern about dual publication or research ethics.

Reviewer #2: In the manuscript by Shin and colleagues human induced pluripotent stem cells were produced and were differentiated into fibroblasts and keratinocytes. Those cells were then used to establish 3D skin equivalents. Establishing high trough put in vitro skin models is important to meet the high demand of new models needed to replace and reduce animal experiments, especially with regard to chemical testing.

The manuscript is nicely written follows a clear path presenting results, although there are some major draw backs

1. In General, the writing is not clear, especially in the methods section. The English should be edited by a native speaker.

2. Methods: Generation of hi-PSC should be explained in more detail.

3. Formation of 3D SKE: Although there is a good separation into dermis and epidermis, the epidermis is not differentiated properly. There is no stratum granulosum and no stratum corneum formed. Differences in stratum basale and spinosum in H&E staining are only minor. Skin models should be improved so that sublayers of the epidermis form properly as seen in 3D SKE made from primary skin cells. Have different timings been tested? 30 days are quite long compared to other published 3D skin models.

4. Further, proper epidermal differentiation should not only be assessed by H&H staining but also IHC, staining for markers specific for certain sublayers, e.g. Fillagrin, Involucrin, Ki-67 or different keratins.

6. PLOS authors have the option to publish the peer review history of their article (what does this mean? ). If published, this will include your full peer review and any attached files.

**Do you want your identity to be public for this peer review?** For information about this choice, including consent withdrawal, please see our Privacy Policy .

Reviewer #1: No

Reviewer #2: No

---

## [Author Response · Author response to Decision Letter 1]

26 Jun 2025

PONE-D-25-01505

Skin Irritation Testing using Human iPSCs derived 3D Skin Equivalent Model

PLOS ONE

Dear Dr. Chung,

Thank you for submitting your manuscript to PLOS ONE. After careful consideration, we feel that it has merit but does not fully meet PLOS ONE’s publication criteria as it currently stands. Therefore, we invite you to submit a revised version of the manuscript that addresses the points raised during the review process.

1) I am not confident that we have a good quality of 3D skin. As pointed out by one of the reviewers, "There is no stratum granulosum and no stratum corneum formed. Differences in stratum basale and spinosum in H&E staining are only minor";

2) Please, see the comments raised by both the reviewers.

We look forward to receiving your revised manuscript.

Kind regards,

Paulo Lee Ho, Ph.D.

Academic Editor

PLOS ONE

Response to Editor:

We sincerely thank the Editor’s genuine interest in our study and are grateful for permitting this revision of our manuscript. We did our best to experiment and consider in response to reviewers' comments. We hope that this revision coupled with our responses, are able to sufficiently address the critiques so our information and experience may be shared with the readership of PLOS ONE. 

Correction part

<Manuscript>

1) Title page

1. Excluding author ‘Se Eun Kim’ and ‘Yun Hyeong Lee’.

2) Abstract

1. Research on artificial skin has advanced significantly, supporting cosmetic ingredient evaluation and skin regeneration treatment development. Artificial skin-based tests are well established, with standardized guidelines and widespread use of three-dimensional (3D) skin equivalent models (SKE) constructed from human primary skin cells. However, in the case of primary cells, there are problems with supply due to limited patients and difficulties in genetic-specific research. To address these issues, recent efforts have focused on differentiating skin cells from human-induced pluripotent stem cells (hiPSCs). � Artificial skin models have emerged as valuable tools for evaluating cosmetic ingredients and developing treatments for skin regeneration. Among them, 3D skin equivalent models (SKEs) using human primary skin cells are widely utilized and supported by standardized testing guidelines. However, primary cells face limitations such as restricted donor availability and challenges in conducting genotype-specific studies. To overcome these issues, recent approaches have focused on differentiating skin cells from human-induced pluripotent stem cells (hiPSCs).

2. The resulting hiPSC-SKE model replicated the primary layers of native skin morphologically, as demonstrated by hematoxylin and eosin staining, and expressed specific markers characteristic of each layer. Furthermore, treatment of hiPSC-SKEs with Triton X-100, a positive control for skin irritation testing, caused severe epidermal cell damage and reduced cell viability. These findings suggest that hiPSC-SKE serves as a promising alternative for skin-related applications, including animal testing replacement and genetic skin disease modeling. � Histological analysis with hematoxylin and eosin staining confirmed that the hiPSC-SKE recapitulated the layered architecture of native human skin and expressed appropriate epidermal and dermal markers. Moreover, exposure to Triton X-100, a known skin irritant, led to marked epidermal damage and significantly reduced cell viability, validating the model’s functional responsiveness. These findings indicate that the hiPSC-SKE model represents a promising alternative for various skin-related applications, including the replacement of animal testing.

3) Introduction

1. The skin is the primary tissue that exists on the body's exterior and performs various physiological functions in response to external environmental exposure (1, 2). � The skin, as the body's outermost tissue, serves as a barrier against environmental stimuli and performs essential physiological functions (1, 2).

2. Numerous studies have focused on creating human 3D skin equivalents (SKEs) using primary skin cells to replicate human skin in vitro (3-5). One advantage of SKEs is their relatively simple production through insert-based methods (6). In addition, when cultured keratinocytes on the surface in contact with the air forms epidermal-like layer, it forms an environment and structural complexity similar to human skin. � To replicate skin in vitro, 3D skin equivalent models (SKEs) have been developed using primary skin cells (3-5). SKEs can be fabricated relatively easily using insert-based methods (6). When keratinocytes are cultured on a surface exposed to air, they form the epidermis-like layer, creating an environment and structural complexity similar to that of human skin.

3. As a result, SKE has been used in various experiments related to skin drug toxicity evaluation, disease modeling, and transplantation therapeutics as wound models (5). � Due to these features, SKEs are widely used in studies related to drug toxicity, disease modeling, and wound healing applications (5).

4. Among the various applications of SKEs, skin irritation and corrosion testing are prominent (7-9). International guidelines for skin irritation testing are well-established, with the Organization for Economic Cooperation and Development test guideline 439 (OECD TG 439) serving as a key reference (7-9). This testing is critical for verifying the safety of products such as cosmetics, pharmaceuticals, and chemicals by evaluating whether a substance causes physical or chemical irritation or damage to the skin. And OECD TG 439 proposes a method that uses an artificial skin model to replace traditional animal testing. � SKEs are widely used for skin irritation and corrosion testing, guided by international standards like the Organization for Economic Cooperation and Development test guideline 439 (OECD TG 439). This testing is crucial for evaluating the safety of cosmetics, pharmaceuticals, and chemicals, and OECD TG 439 promotes artificial skin models as a replacement for animal testing (7-9).

5. Artificial skin used according to these guidelines typically includes Reconstructed Human Epidermis (such as RhE; EpiskinTM, EpiDermTM, SkinEthicTM RHE) and Dermo-Epidermal SKEs created from commercial primary skin cells (7-9). In addition to safety evaluations, keratinocytes and fibroblasts derived from human tissues are commonly used in research on skin diseases and therapeutic agents (6, 12-15). While primary cells reflect in vivo maturity, challenges such as limited supply, high cost, and senescence from passaging remain. Additionally, simulating genetic disease models is difficult due to limitations in manufacturing patient-specific products, and immune rejection complicates therapeutic agent transplantation. To address these issues, human induced pluripotent stem cells (hiPSCs), known for their self-renewal and pluripotency, have emerged as a promising source (16, 17). The advantages of hiPSC-derived cells include: 1) overcoming cell supply constraints through self-renewal, 2) disease modeling using patient-derived genetic samples or gene editing, and 3) superior therapeutic effects through greater engraftment with low immunogenicity (16, 17). Some studies have successfully produced 3D SKEs from hiPSC-derived skin cells (18-21). However, reports on the production and use of hiPSC-derived SKEs (hiPSC-SKEs) are still limited, and creating a comprehensive database of studies is essential to refine these models. In this study, we established a protocol for producing skin cells from hiPSCs and successfully generated hiPSC-SKEs. Morphological analysis confirmed that hiPSC-SKEs replicate human skin characteristics, and skin irritation tests using Triton X-100 were successfully conducted. These databases are expected to play a significant role in fields such as medicine, cosmetics, and drug development, as hiPSC-SKEs evolve into more advanced skin models in the future. � Artificial skin models used according to established guidelines typically include reconstructed human epidermis (RhE) products such as Episkin™, EpiDerm™, and SkinEthic™ RHE, as well as dermo-epidermal SKEs constructed from commercially available primary skin cells  (7-9). In addition to safety evaluations, skin cells are commonly used in research on skin diseases and therapeutic agents (6, 12-15). While primary cells reflect in vivo maturity, their use is limited by supply shortages, high costs, and senescence during passaging. Additionally, modeling genetic skin diseases is challenging due to difficulties in producing patient-specific products, and immune rejection complicates therapeutic transplantation. To address these issues, human induced pluripotent stem cells (hiPSCs) have emerged as a promising source (16, 17). Advantages of hiPSC-derived cells include: (1) unlimited supply through self-renewal, (2) disease modeling via patient-specific genetics or gene editing, and (3) improved therapeutic potential with better engraftment and low immunogenicity (16, 17). Several studies have successfully generated 3D SKEs from hiPSC-derived skin cells (18-21). However, reports on hiPSC-derived SKEs (hiPSC-SKEs) remain limited, and a comprehensive study database is needed to optimize these models. In this study, we established a protocol to differentiate skin cells from hiPSCs and generated hiPSC-SKEs. Morphological analyses confirmed that hiPSC-SKEs replicate human skin features, and skin irritation tests using Triton X-100 were successfully performed. These datasets are expected to contribute significantly to cosmetics, chemicals, and drug development as hiPSC-SKEs advance as skin models.

4) Materials & Methods

1. Added sentence ‘(Generation of human induced pluripotent stem cells (hiPSCs)) 1.5 × 10⁶ human dermal fibroblasts (BJ cells; ATCC, VA, USA) were transfected with episomal vectors (pCLXE-hOCT4/p53, pCLXE-hSOX2/KLF4, and pCLXE-hL-MYC/LIN28A; Addgene, MA, USA) using the P2 Primary Cell 4D-Nucleofector Kit (Lonza, Basel, Switzerland) according to the manufacturer’s protocol (22, 23). The mixture of cells and vectors was transferred into a Nucleocuvette Vessel and subjected to electroporation. Transfected cells were plated onto 1:100 Matrigel (Corning, NY, USA)-coated 6-well plates and cultured for 2 days in Dulbecco’s Modified Eagle’s Medium (DMEM; Gibco, NY, USA) supplemented with 10% fetal bovine serum (FBS; Sigma-Aldrich, St. Louis, MO, USA), 0.1 mM non-essential amino acids (NEAA; Gibco), and 1% penicillin/streptomycin (P/S; Gibco). After 2 days, the medium was replaced with TeSR-E7 medium (STEMCELL Technologies, Vancouver, Canada), which was changed daily. Cells were cultured for an additional 10–14 days until PSC-like colonies appeared. After picking the PSC-like colonies, they were seeded onto Matrigel-coated 4-well plates in mTeSR1 (STEMCELL Technologies Inc; Vancouver, BC, Canada) medium supplemented with 10 µM Y-27632 (Tocris Bioscience, Bristol, UK). The medium was changed daily with fresh mTeSR1 during subsequent culture.’.

2. hiPSCs were produced using a pluripotency-related reprogramming episomal vectors (addgene, MA, USA) in human dermal fibroblasts (ATCC, VA, USA) (22, 23). hiPSCs were seeded on 60 mm cell culture dish coated with 1:100 Matrigel (Corning Inc.; Corning, NY, USA) and cultured by replacing the mTeSR1 (STEMCELL Technologies Inc; Vancouver, BC, Canada) medium every day. After 3 days, when 90-100% confluency was achieved, hiPSCs were detached using DPBS (welgene, Gyeongsan, South Korea) containing 0.5 mM EDTA (Gibco, NY, USA). Detached hiPSCs were collected in 6 ml mTeSR1 containing 10 µM Y-27632 (Tocris Bioscience, Bristol, UK) and placed in a 60 mm petri dish coated with F127 (Sigma-Aldrich, St. Louis, USA). Afterwards, they were cultured for one day in a shaker set at 65-70 rpm in a 37 °C, 5% CO2 incubator. The next day, embryonic bodies (EBs) formation was confirmed and mTeSR1 was replaced every day for 2 days (D3). 150-200 EBs were seeded on Matrigel-coated 60 mm cell culture dishes using fibroblast differentiation media 1 (FDM1) and replaced for 3 days. FDM1 is a 3:1 mixture of DMEM (Gibco, NY, USA) and F12 (Gibco) supplemented with 5% fetal bovine serum (FBS; Sigma-Aldrich), 5 µg/mL Insulin (Sigma-Aldrich) and 10 ng/ml epithelial growth factor (EGF; PeproTech, NJ, USA). After 3 days, FDM2 was replaced and cultured for 3 days. FDM2 is a medium supplemented with 25 ng/ml bone morphogenic protein 4 (BMP4; Peprotech) to FDM1. Next, it was cultured for 7 days after replacing with FDM3. FDM3 is a 1:1 mixture of DMEM and F12 supplemented with 1% non-essential amino acids solution (NEAA; Gibco). After 7 days, it was replaced with FDM1 medium again and cultured for 4 days. When it reached 70-80% confluency, it was detached using a 5:1 mixture of TrypLE (Gibco) and 0.25% Trypsin/EDTA (T/E; Gibco) (D20). The detached cells were seeded on matrigel-coated cell culture dishes with FDM1 medium and maintained in culture until D30 (Fig. 1A). � hiPSCs were seeded onto 1:100 Matrigel-coated 60 mm cell culture dishes and maintained in mTeSR1 medium with daily medium changes. Upon reaching 90–100% confluency after 3 days, the cells were gently dissociated using DPBS (Welgene, Gyeongsan, South Korea) containing 0.5 mM EDTA (Gibco). The detached hiPSCs were resuspended in 6 ml mTeSR1 supplemented with 10 µM Y-27632 and transferred to F127-coated (Sigma-Aldrich) 60 mm Petri dishes to initiate embryoid body (EB) formation (D0). Cells were cultured on a shaker at 65–70 rpm in a 37 °C incubator with 5% CO₂ for 24 hr. On the following day (D1), EB formation was confirmed, and mTeSR1 was replaced daily for two additional days (until D3). Approximately 150–200 EBs were then transferred onto Matrigel-coated 60 mm culture dishes and cultured in fibroblast differentiation medium 1 (FDM1) for 3 days (D3-D6). FDM1 consisted of a 3:1 mixture of DMEM (Gibco) and F12 (Gibco), supplemented with 5% FBS, 5 µg/ml insulin (Sigma-Aldrich), and 10 ng/ml epidermal growth factor (EGF; PeproTech, NJ, USA). After 3 days, the medium was changed to fibroblast differentiation medium 2 (FDM2), which is FDM1 supplemented with 25 ng/ml bone morphogenetic protein 4 (BMP4; PeproTech), and cultured for another 3 days (D6-D9). Subsequently, the cells were cultured in fibroblast differentiation medium 3 (FDM3) for 7 days (D9-D16). FDM3 consisted of a 1:1 mixture of DMEM and F12 supplemented with 5% FBS

---

## [Decision Letter · Decision Letter 1]

30 Jul 2025

Skin Irritation Testing using Human iPSCs derived 3D Skin Equivalent Model

PONE-D-25-01505R1

Dear Dr. Chung,

We’re pleased to inform you that your manuscript has been judged scientifically suitable for publication and will be formally accepted for publication once it meets all outstanding technical requirements.

Kind regards,

Paulo Lee Ho, Ph.D.

Academic Editor

PLOS ONE

Additional Editor Comments (optional):

Reviewers' comments:

Reviewer's Responses to Questions

**Comments to the Author**

1. If the authors have adequately addressed your comments raised in a previous round of review and you feel that this manuscript is now acceptable for publication, you may indicate that here to bypass the “Comments to the Author” section, enter your conflict of interest statement in the “Confidential to Editor” section, and submit your "Accept" recommendation.

Reviewer #1: All comments have been addressed

Reviewer #2: All comments have been addressed

2. Is the manuscript technically sound, and do the data support the conclusions?

Reviewer #1: Yes

Reviewer #2: Yes

3. Has the statistical analysis been performed appropriately and rigorously? 

Reviewer #1: Yes

Reviewer #2: Yes

4. Have the authors made all data underlying the findings in their manuscript fully available?

Reviewer #1: Yes

Reviewer #2: Yes

5. Is the manuscript presented in an intelligible fashion and written in standard English?

Reviewer #1: Yes

Reviewer #2: Yes

6. Review Comments to the Author

Reviewer #1: The focus on methodological detail, quantitative rigor, and manuscript clarity is appropriate and consistent with standards for publication.

1. General response to reviewer

Could be slightly more specific about where/how improvements were made to increase the credibility of the response.

2. Cell purity and characterization

Provides new data and clearly explains the methods used to address the concern. Supplementary figure addition is appropriate.

3. 3D model construction and media rationale

They explained both the rationale and optimization efforts. The inclusion of comparisons to past protocols strengthens the response.

4. Triton X-100 rationale

Clear explanation grounded in OECD guidelines. Appropriately referenced standard practice.

5. Statistical analysis

They addressed the concern well by explicitly adding the use of Tukey’s test. Including the method directly in the manuscript improvede transparency.

6. Mechanism of Triton X-100

They contextualized the choice and clarified the scope of the study.

7. Methods section clarity

They directly addressed with added detail. This strengthens reproducibility.

8. hiPSC generation protocol

Fully addressed with specifics on episomal vectors, timing, and selection methods.

9. Differentiation protocols

Providing both text and schematic (Figure 1) was an effective way to present complex information.

10. hiPSC-SKE construction

Well-clarified with all relevant parameters now included (collagen type, seeding density, medium details).

11. Immunofluorescence

Adequately addressed with catalog numbers, companies, and DAPI info.

12. Results section improvements

Strong improvement by adding quantification (cell marker % and thickness). Reinforces the conclusions with numbers.

13. Discussion section: comparisons and limitations

They’ve acknowledged model limitations, compared to commercial standards, and outlined future directions clearly.

Overall comment

The revision is highly suitable and demonstrates a robust scientific response. The authors addressed their concerns with specific data and methodological transparency, and significantly improved the manuscript clarity, rigor, and completeness. The integration of supplemental figures, quantitative analysis, and expanded methodological descriptions shows strong scientific maturity.

Minor improvements could still be made by weaving more of the changes directly into the responses (e.g., quoting or referencing the exact location in the revised manuscript), but overall, the quality is good.

Reviewer #2: Although epidermal differentiation is still not perfekt, those skins are a valuable tool for toxicity testing.

7. PLOS authors have the option to publish the peer review history of their article (what does this mean? ). If published, this will include your full peer review and any attached files.

**Do you want your identity to be public for this peer review?** For information about this choice, including consent withdrawal, please see our Privacy Policy .

Reviewer #1: No

Reviewer #2: No

---

## [Editor Report · Acceptance letter]

PONE-D-25-01505R1

PLOS ONE

Dear Dr. Chung,

I'm pleased to inform you that your manuscript has been deemed suitable for publication in PLOS ONE. Congratulations! Your manuscript is now being handed over to our production team.

Kind regards,

on behalf of

Dr. Paulo Lee Ho

Academic Editor

PLOS ONE